# Human Sex Matters: Y-Linked Lysine Demethylase 5D Drives Accelerated Male Craniofacial Osteogenic Differentiation

**DOI:** 10.3390/cells11050823

**Published:** 2022-02-26

**Authors:** Madlen Merten, Johannes F. W. Greiner, Tarek Niemann, Meike Grosse Venhaus, Daniel Kronenberg, Richard Stange, Dirk Wähnert, Christian Kaltschmidt, Thomas Vordemvenne, Barbara Kaltschmidt

**Affiliations:** 1Molecular Neurobiology, Bielefeld University, Universitätsstrasse 25, 33615 Bielefeld, Germany; madlen.merten@arcor.de (M.M.); tarek.niemann@uni-bielefeld.de (T.N.); meike.grosse_venhaus@uni-bielefeld.de (M.G.V.); 2Department of Cell Biology, Bielefeld University, Universitätsstrasse 25, 33615 Bielefeld, Germany; johannes.greiner@uni-bielefeld.de (J.F.W.G.); c.kaltschmidt@uni-bielefeld.de (C.K.); 3Forschungsverbund BioMedizin Bielefeld FBMB e.V., Maraweg 21, 33617 Bielefeld, Germany; dirk.waehnert@evkb.de (D.W.); thomas.vordemvenne@evkb.de (T.V.); 4Department of Regenerative Musculoskeletal Medicine, Institute for Musculoskeletal Medicine, University Hospital Muenster, Westfaelische Wilhelms University Muenster, Albert-Schweitzer-Campus 1, Building D3, 48149 Muenster, Germany; daniel.kronenberg@ukmuenster.de (D.K.); richard.stange@ukmuenster.de (R.S.); 5Department of Trauma and Orthopedic Surgery, Protestant Hospital of Bethel Foundation, University Hosptal OWL of Bielefeld University, Campus Bielefeld-Bethel, Burgsteig 13, 33617 Bielefeld, Germany

**Keywords:** adult human stem cells, sexual dimorphisms, osteogenic differentiation, KDM5D, calvarial bone regeneration, transcriptional profiling, KDOAM-25

## Abstract

Female sex is increasingly associated with a loss of bone mass during aging and an increased risk of developing nonunion fractures. Hormonal factors and cell-intrinsic mechanisms are suggested to drive these sexual dimorphisms, although underlying molecular mechanisms are still a matter of debate. Here, we observed a decreased capacity of calvarial bone recovery in female rats and a profound sexually dimorphic osteogenic differentiation in human adult neural crest-derived stem cells (NCSCs). Next to an elevated expression of pro-osteogenic regulators, global transcriptomics revealed Lysine Demethylase 5D (KDM5D) to be highly upregulated in differentiating male NCSCs. Loss of function by siRNA or pharmacological inhibition of KDM5D significantly reduced the osteogenic differentiation capacity of male NCSCs. In summary, we demonstrated craniofacial osteogenic differentiation to be sexually dimorphic with the expression of KDM5D as a prerequisite for accelerated male osteogenic differentiation, emphasizing the analysis of sex-specific differences as a crucial parameter for treating bone defects.

## 1. Introduction

Sexual dimorphisms are increasingly noticed to account for the severity of human disease phenotypes [1,2]. In bone regeneration, the severity and occurrence of bone disorders is linked to the sex of an individual [3], reviewed in [4]. Here, female sex in particular is linked to a gradual loss of bone mass after the age of around 35 [5,6] and considered as a major risk factor for decreased osseointegration of implants [7] and the development of nonunion fractures [3,8]. Postmenopausal women also show an increased prevalence of osteoporosis compared to their male counterparts [9,10,11]. While aggressive osteoclastic bone resorption was recently reported to be linked to the age and menopausal status of female individuals [6], the bone formation rate simultaneously decreases [6,11]. At a mechanistic level, estrogen was reported to regulate bone resorption and formation [12] and the osteogenic activity of stem cells in mouse osteoporosis models [13]. However, Meszaros and colleagues reported a significantly increased regeneration of cranial bone defects in castrated or non-operated male mice compared to non-operated or ovariectomized female animals, suggesting the additional presence of non-hormonal mechanisms guiding sexual dimorphic bone regeneration rates [14]. In this line, increasing evidence suggests that stem cell-intrinsic mechanisms account for sex-specific differences in bone recovery and the occurrence of bone disorders (reviewed in [4]). For instance, male mouse skeletal muscle-derived stem cells were reported to reveal a greater amount of mineralization and ALP-activity after osteogenic differentiation compared to their female counterparts [15]. In humans, male adipose tissue stem cells (ASCs) were likewise shown to more efficiently give rise to osteogenic derivates compared to female ASCs [16]. Despite these promising findings, the molecular mechanisms of the observed sex-specific differences in the osteogenic differentiation of stem cells, particularly in the cranial and craniofacial region, are still a matter of debate.

Facing this challenge, this study demonstrated sex-specific differences in the regeneration of critical-size calvarial rat bone defects as well as in the osteogenic differentiation of adult human craniofacial stem cells and determined the underlying molecular drivers regulating sex-specific stem cell behavior. We particularly utilized adult human neural crest-derived stem cells (NCSCs) from the nasal cavity as a human in vitro model for investigating sexual dimorphisms in craniofacial osteogenic differentiation in accordance with our previous studies [17,18,19]. During embryonic development, neural crest cells give rise to the majority of the craniofacial skeleton [20,21,22,23,24], emphasizing the suitability of adult NCSCs as cellular model systems for human craniofacial bone regeneration. Accordingly, adult NCSCs were discussed as potent drivers of craniomaxillofacial tissue repair [25], for instance in oral surgery [26], and were broadly described to give rise to mineralizing osteoblasts in vitro [19,27,28]. In the present study, collagen type I fibers harboring 30 nm pores were applied as a pro-osteogenic biophysical cue to guide osteogenic differentiation of NCSCs [18]. Next to collagen type I fibers, the clinically approved hemostatic collagen sponge Spongostan was utilized for investigating the sexual dimorphic regeneration of critical-size calvarial rat bone defects. We recently demonstrated the distinct nano- and microtopography of Spongostan to be solely sufficient for guiding bone recovery in critical-size calvarial bone defects in male rats [29,30]. Taking advantage of these biophysical cues driving osteogenic differentiation, we found in the current study an elevated regeneration of critical-size calvarial bone defects in male rats after transplantation of collagen fibers or Spongostan compared to their female counterparts. In accordance with our observations in vivo, human NCSCs from female individuals revealed a strong delay in osteogenic differentiation on collagen type I fibers over time in comparison to NCSCs from male donors. Global gene expression profiling following osteogenic differentiation revealed a sex-specific expression pattern of negative osteogenic regulators such as micro RNA 137 (MIR137HG) and taxilin gamma (TXLNG) in female NCSCs, while pro-osteogenic regulators including C-FOS and Cytochrome P450 Family 7 Subfamily B Member 1 (CYP7B1) were upregulated in male stem cells. We further functionally validated the expression of the Y-linked Lysine Demethylase 5D (KDM5D) as a relevant driver of the observed elevated osteogenic differentiation capacity of male craniofacial NCSCs.

## 2. Materials and Methods

### 2.1. Animals

This study was approved by the responsible authority (LANUV, Approval Ref. No. 81-0204.2018.A188). All animal work was performed in accordance with the policies and procedures established by the Animal Welfare act, the National Institutes of Health Guide for Care and Use of Laboratory Animals, and the National Animal Welfare Guidelines. For this study, a minimum number of 5 male and 5 female white Wistar rats (average weight of 300 g at day of surgery) from our own breeding at the University of Bielefeld were randomly assigned to either group A (empty control, n = 5), group B (collagen fibers, n = 5), group C (Spongostan, n = 5). All animals were housed 2-per-cage under a 12 h light/dark cycle with food and water provided ad libitum. All efforts were made to minimize potential suffering in accordance with the 3R (replacement, reduction, refinement) guidelines.

### 2.2. Animal Surgery and Postoperative Care

Male rats were anesthetized via intraperitoneal injection of ketamine hydrochloride (60 mg/kg, Inresa Arzneimittel GmbH, Freiburg, Germany) and medetomidine (0,3 mg/kg, Dechra Veterinary Products Ltd., Shrewsbury, UK) after subcutaneous injection of tramadol (20 mg/kg, Grünenthal GmbH, Aachen, Germany) as previously described [29]. Female rats were administered an intraperitoneal injection of “Hellabrunner Mischung” comprising ketamine hydrochloride/xylazine hydrochloride solution (1 mL/kg, Sigma Aldrich, Taufkirchen, Germany). The surgical procedure followed the description of Spicer and colleagues [31]. Briefly, the fur was shaved and the area of surgery was cleaned using an iodine swab. A skin incision, followed by periosteum removal, was made. Afterwards, on both sides the trepanation was performed using a 5 mm hollow drillhead (Trephines 229 RAL 040: Hager and Meisinger GmbH, Neuss, Germany) and a surgical dental drill (Implantmed: W&H Dentalwerk Bürmoos GmbH, Bürmoos, Austria) with a constant speed of 2000 rpm. The circular bone fragment was lifted out, the defect was washed with sterile saline solution and the holes were treated according to the group the animal was assigned to as follows: control without any filling (group A), collagen fibers (group B) or Spongostan (group C). For group B, collagen type I fibers were prepared as described previously [18]. Briefly, tendons were cut into small pieces and incubated for 2 h at 37 °C in laundry detergent (Persil Megaperls; Henkel AG, Düsseldorf, Germany), followed by thorough washing and decomposition into fibrils using a liquid nitrogen-cooled mortar and subsequent ultrasonic disintegration. Pure collagen fibrils were collected by centrifugation and checked by high-power phase contrast microscopy prior to transplantation. For group C, Spongostan (Ferrosan Medical Devices, Søborg, Denmark; marketed by Ethicon Biosurgery, Johnson and Johnson, New Brunswick, NJ, USA) was commercially purchased, cut into cubic sections (1 mm^3^) and pre-wetted in 0.9% NaCl solution (B. Braun Melsungen AG, Melsungen, Germany) for 30 min prior to transplantation [29]. Thereafter, the periosteum was closed with sutures (Prolene 5-0, Ethicon, Johnson and Johnson, New Brunswick, NJ, USA) and the skin was closed by four single back-and-forth sutures (Prolene 3-0, Ethicon, Johnson and Johnson, New Brunswick, NJ, USA). After surgery, the rats were placed in a warmed incubator and housed singly for 3 days.

### 2.3. Euthanasia and Sample Extraction

Animals were euthanized thirty days postoperatively using the Exposure Line carbon dioxide box bioscope (Ehret, Freiburg, Germany). Calvaria were extracted by removing soft tissue and surrounding bone, followed by fixation using 4% paraformaldehyde (PFA) and storage at 4 °C.

### 2.4. Micro-CT

All samples underwent micro-computed tomography as previously described [29] using a SkySkan 1176 (Bruker, Kontich, Belgium).

### 2.5. Histology

Extracted calvaria were fixed with 4% paraformaldehyde (PFA), subsequently decalcified using ethylenediaminetetraacetic acid (EDTA) and embedded in paraffin. Serial sections were stained with Goldner’s trichrome stain, while nuclei were stained with hematein followed by microscopical examination.

### 2.6. Osteogenic Differentiation of Human Neural Crest Derived Inferior Turbinate Stem Cells

Human inferior turbinates were obtained during routine surgery after an informed consent process, according to local and international guidelines (Bezirksregierung Detmold, Germany). NCSCs (inferior turbinate stem cells, ITSCs) were isolated via cultivation in DMEM F12 supplemented with 40 ng/mL Fibroblast Growth Factor (FGF)-2, 20 ng/mL Epidermal Growth Factor (EGF), B27 supplement (hereafter referred to as stem cell medium) and 2 mg/mL Heparin (Sigma-Aldrich) to allow the formation of primary spheres as previously described [17]. Isolated NCSCs were cultivated in stem cell medium with an additional 10 % human blood plasma according to [32]. All experimental procedures were ethically approved by the ethics board of the medical faculty of the University of Münster (No. 2012–015-f-S). The ages of the donors at the time of surgery are depicted in Table 1.

Osteogenic differentiation was induced topologically using collagen fibers according to our previously reported protocol [18]. Briefly, NCSC-populations from three male and three female donors (for passages, see Table 1) precultivated in stem cell medium with 10% human blood plasma were seeded onto collagen fibers endogenously harboring 30 nm pores and cultivated in DMEM supplemented with 10% fetal calf serum at 37 °C and 5% CO_2_ in a humidified incubator (Binder, Tuttlingen, Germany). Calcium deposition was visualized by Alizarin Red staining and imaging. Alizarin Red was extracted using acetic acid and treated with ammonium hydroxide for neutralization followed by quantification via photometric measurement at 405 nm as shown in [33].

### 2.7. RNA Isolation and Sequencing

NCSC-populations from three male and three female donors (see Table 1) were osteogenically differentiated for 14 and 30 days as described above, followed by RNA isolation with a NucleoSpin RNA kit (Macherey-Nagel, Düren, Germany) according to the manufacturer’s guidelines. The quality and quantity of the isolated RNA were determined via NanoDrop (Thermo Fisher Scientific,Waltham, MA, USA). Library preparation using the NEB Next Ultra Library Prep Kit (New England Biolabs, Frankfurt am Main, Germany) and sequencing on the Illumina Novaseq6000 platform were carried out by Novogene (Cambridge, UK), followed by bioinformatic analysis. RNA seq was performed with an average of 43,196,219 raw reads per sample. Hisat2 was used for mapping to the reference genome GRCh38/hg19. Quantification of the gene expression was analyzed via HTSeq and, for differential gene expression analysis, the DESeq2 R Package was used [34,35]. Enrichment analysis of differentially expressed coding genes was conducted for GO Enrichment via the R Packages GOSeq, topGO and hmmscan and analysis of Kyoto Encyclopedia of Genes and Genomes (KEGG) pathway enrichment was performed using KOBAS and PANTHER.

### 2.8. Generation of Lentiviral Vectors

ShRNAs against KDM5D were commercially purchased (MISSION shRNA Bacterial Glycerol Stock, clone IDs: NM_004653.3-4909s21c1, NM_004653.2-4787s1c1, NM_004653.3-1368s21c1, NM_004653.2-3246s1c1; Sigma Aldrich). Target sequences are depicted in Figure 5A. Lentiviral vectors were generated following the protocol of Tang and colleagues [36] using transient cotransfection of HEK 293T cells with a three-plasmid system. Lentiviral transfer vector plasmid, packaging plasmid and envelope plasmid were mixed and 1.25 mM PEI were added and incubated for 10 min at room temperature. Meanwhile, HEK 293T cells were trypsinized, washed twice with 1 × PBS and 2 × 10^6^ cells were resuspended in media. The vector/PEI mixture was added to the cells and immediately plated onto a plate and incubated at 37 °C and 5% CO_2_ in a humidified incubator. After 48 h, the supernatant containing lentiviral vectors was collected and centrifugated at 3000 rpm for 5 min at room temperature followed by filtration through a 0.45 µm mesh. Virus was transferred into conical tubes and spun at 20,000 rpm for 2 h at 4 °C in a Beckman TST2838 swinging bucket rotor. After spinning, the supernatant was discarded and the virus was resuspended in a desired volume of serum-free media and stored at −80 °C.

### 2.9. siRNA-Mediated Knockdown and Pharmacological Inhibition of KDM5D

NCSCs were transduced with lentivirally packaged vectors produced as described above followed by verification of knockdown using qPCR (see below). Knockdown cells were seeded on collagen type I fibers and cultivated for 30 days as described above. Pharmacological inhibition of KDM5D was performed by exposing NCSCs to 1 µM, 10 µM and 100 µM KDOAM-25 (KDOAM hydrochloride hydrate, SML2774, Sigma Aldrich) for 30 days of osteogenic differentiation on collagen type I fibers as described above.

### 2.10. Western Blot

KDM5D Knockdown (KD) and Wildtype (WT) ITSCs were harvested by surface scraping followed by cell lysis on ice with lysis buffer (10 mM Tris, 1% SDS, 3 mM EDTA). Protein concentrations of the samples were adjusted to 20 µg total protein amount and mixed with 4× loading buffer followed by heating at 95 °C for 8 min. Samples were subjected to electrophoresis on 10% denaturing SDS polyacrylamide gel and transferred with a semi-dry blotter to a nitrocellulose membrane (Carl Roth GmbH, Karlsruhe, Germany). Blocking of membrane with 5% milk powder (Carl Roth GmbH, Karlsruhe, Germany) in 1× TBS with 0.05% Tween 20 (Sigma-Aldrich, Taufkirchen, Germany) was followed by incubation with the first antibody against JARID1D/KDM5D (1:500; Cell Signaling Technology Inc., Danvers, MA, USA) in 1× TBS with 5% milk powder and 0.05% Tween 20 while shaking at 4 °C overnight. HRP-linked secondary antibody (1:4000; Cell Signaling Technology Inc.) was applied for 1h at RT. Visualization was performed via enhanced chemiluminescence. Beta-actin antibody (1:2000; Cell Signaling Technology Inc.) was applied as control for 1 h at RT followed by exposure to secondary antibody as described above.

### 2.11. qPCR

Total RNA was isolated as described above followed by cDNA synthesis using the First Strand cDNA Synthesis Kit (Fermentas, St. Leon-Rot, Germany). qPCR was performed using the Platinum SYBR Green qPCR Super-Mix UDG (Invitrogen, Life Technologies GmbH, Darmstadt, Germany) according to the manufacturers guidelines and assayed with a Rotor Gene 6000 (QIAGEN, Hilden, Germany). The primer sequences were: GGCTGAGTCTTTTGACACCTGG (KDM5D forward), CAGGCAGTTCTTCAGTCGCTGA (KDM5D reverse), CTGCACCACCAACTGCTTAG (GAPDH forward), GTCTTCTGGGTGGCAGTGAT (GAPDH reverse), TGGGCAAGAACACCATGATG (RPL0 forward),AGTTTCTCCAGAGCTGGGTTGT (RPL0 reverse); qPCR was conducted with three replicates and normalized to GAPDH (analysis depicted in Figure 5) or RPL0 (analysis depicted in Appendix A), and the relative expression ratio was calculated as 2^−(Ct target gene−Ct reference gene)^.

### 2.12. Statistical Analysis

Quantification of Alizarin Red S was performed with three replicates and statistically analyzed using the Mann–Whitney test utilizing the Graph Pad Prism software (GraphPad Software, Version 5.02, 2008, La Jolla, CA, USA). qPCR data obtained with three replicates (see chapter qPCR above) were likewise statistically analyzed using the Mann–Whitney test with Graph Pad Prism software (GraphPad Software). * *p* < 0.05 was considered significant.

## 3. Results

### 3.1. Calvarial Critical-Size Defects in Female Rats Showed Impaired Regeneration Compared to Male Animals

We determined potential differences in bone recovery between the sexes by comparing the regeneration of critical-size calvarial bone defects in male and female rats. To allow recovery of the lesion, we applied the collagen type I fibers or the collagen sponge Spongostan in accordance with our previous observations [29,30] after the trepanation of calvarial bones. Although no closures of the critical-size defects were observable in controls via micro-computed tomography (µCT), we observed a significantly reduced bone mineral density (BMD) in female rats compared to male ones (Figure 1A). A significantly decreased bone volume (BV) and BMD was further observable in female animals compared to their male counterparts after transplantation of collagen type I fibers (Figure 1B) or Spongostan (Figure 1C). In particular, a complete closure of the lesion was observable in Spongostan-treated male rats, while female rats revealed only partial recovery (Figure 1C). Histological examination of the newly formed bone confirmed the observations made via µCT, particularly revealing increased amounts of newly formed bone in male rats compared to female animals after transplantation of Spongostan or collagen type I fibers (Figure 1D, stars indicate new bone). Thus, we demonstrate here a strong and sex-specific impairment of female rats in regenerating calvarial bone lesions compared to males.

### 3.2. Female Adult Human Craniofacial Stem Cells Showed Strongly Delayed Osteogenic Differentiation

Alongside assessing the sex-specific differences in the bone regeneration of calvarial lesions in vivo, we investigated potential sex-dependent differences in osteogenic differentiation in a human model system. We used human craniofacial NCSCs from both sexes and cultivated them on collagen fibers (Figure 2A) to induce osteogenic differentiation by physical cues only [18]. NCSCs from male individuals revealed a strong Alizarin Red S-stained calcium deposition indicating successful osteogenic differentiation after 30 days of culture (Figure 2B). On the contrary, we observed no signs of Alizarin Red S-positive calcification in female NCSCs cultivated under the same conditions for 30 days (Figure 2C). After 40 additional days (day 70) of culture on collagen type I fibers, NCSCs from female donors were also able to differentiate and showed Alizarin Red S-stained calcium deposits (Appendix A). In summary, adult NCSCs from female individuals differentiated on collagen type I fibers but revealed a strong delay in osteogenic differentiation over time in comparison to male NCSCs (under physiological differentiation cues without hormones).

### 3.3. Transcriptome Analysis Revealed an Extreme Sexually Dimorphic Gene Expression

To determine the molecular basis of their sexually dimorphic differentiation behavior, RNA-sequencing of adult human craniofacial NCSCs from three female and three male donors was performed after 14 and 30 days of osteogenic differentiation. Global gene expression profiling revealed the expression of already known osteogenic markers such as ALP, Runx2 and Osteonectin in male and female NCSCs after 14 and 30 days of differentiation without observable differences between the sexes (Figure 2D,E). We found 507 (14 days) and 887 (30 days) transcripts solely expressed in male NCSCs, while 543 (14 days) and 454 (30 days) were exclusively present in female NCSCs upon differentiation (Figure 2D,E). Interestingly, we observed strong differences in the percentages of genes belonging to the molecular function GO terms “binding” and “molecular function regulator”, which were elevated in male NCSCs compared to female ones after 14 days of differentiation (Figure 2D). On the contrary, the percentages of genes belonging to the molecular function GO-term “transporter activity” was found to be increased in female NCSCs (Figure 2D). We further observed nearly similar amounts of molecular function-related genes after 30 days of differentiation between male and female NCSCs (Figure 2E). Likewise, female and male NCSCs revealed nearly similar percentages of expressed genes belonging to biological process GO terms such as “biological adhesion”, “biological regulation”, “cellular process” or “response to stimulus” (Figure 2D,E). These findings show that female NCSCs, at least to some extent, undergo osteogenic differentiation upon culture on collagen fibers despite the lack of calcium deposition after 30 days.

On the contrary, in male NCSCs, differential gene expression analysis revealed a significantly elevated expression of characteristic osteogenic markers and regulators such as BMP4, MSX2, C-FOS, COL4A4, COL11A1 and ITGA1 (Table 2). In addition to this osteogenic expression pattern, hierarchical clustering of differentially expressed genes (DEGs) showed a cluster of upregulated genes in male NCSCs comprising, among others, the known autosomal osteogenic regulator Cytochrome P450 Family 7 Subfamily B Member 1 (CYP7B1) and the lnRNA family with sequence similarity 157 (FAM157C) (Figure 3A,B). Among others, this cluster further included Y-linked genes such as the lysin demethylase 5D (KDM5D) and the putative osteogenic regulator Ubiquitin Specific Peptidase 9 Y-Linked (USP9Y) (Figure 3A,B). Contrarily, female NCSCs revealed an increased expression of BMP7 (Table 2) as well as a cluster of upregulated genes comprising the pro-osteogenic regulators KDM6A and XIST compared to male after 30 days of differentiation (Figure 3C,D). However, we observed negative regulators of osteogenic differentiation such as the X-linked gene taxilin gamma (TXLNG) (Figure 3C,D) or MIR137HG (Table 2) to be significantly higher expressed in female NCSCs in comparison to their male counterparts.

In addition to the determination of distinct clusters of differentially expressed genes or single osteogenic markers and regulators, we performed Kyoto Encyclopedia of Genes and Genomes (KEGG) pathway and Gene Ontology (GO) term analysis. In accordance with our findings showing differences in the osteogenic differentiation between male and female NCSCs, differentially expressed genes belonged, among others, to the osteogenic-related KEGG pathways “ECM-receptor-interaction”, “cell adhesion molecules” or “focal adhesion”, the “TGF-beta signaling pathway” or the “MAPK signaling pathway” (Figure 4C). Accordingly, differentially expressed genes were annotated to the GO terms “cell adhesion”, “cytoskeleton organization” or “extracellular region” (Figure 4C).

### 3.4. Identification of Y-Linked Lysin Demethylase 5D as a Novel Regulator of Osteogenic Differentiation in Male NCSCs

Among the genes in cluster A (Figure 3A,B) upregulated in male NCSCs undergoing osteogenic differentiation, we found Ribosomal Protein S4 Y-Linked 1 (RPS4Y1) and lysin demethylase 5D (KDM5D, Figure 4A, arrows) to be the most significantly and strongly downregulated in female NCSCs after 14 days of differentiation (Figure 4A). In addition to DEAD-Box Helicase 3 Y-Linked (DDX3Y), the transcripts of RPS4Y1 and KDM5D (Figure 4B, arrows) were likewise the most significantly downregulated in female NCSCs undergoing osteogenic differentiation for 30 days (Figure 4B). While increased transcription of RPS4Y1 might be a potential hint of differences in protein translation, histone methylation and histone demethylases such as KDM4A are discussed to regulate osteogenic differentiation (reviewed in [37]) [38]. Thus, we particularly focused on investigating the regulatory functions of KDM5D in more detail. As a functional validation, we used siRNAs against four distinct loci in the mRNA of KDM5D (Figure 5A) followed by their lentiviral transduction into male NCSCs. Firstly, knockdown of KDM5D was verified by qPCR and Western Blot. Here, qPCR depicted a significantly reduced expression of KDM5D compared to untreated controls and GFP-transduced controls in undifferentiated NCSCs (Figure 5B). Accordingly, a strongly reduced protein amount of KDM5D (190 kDa) was observable in knockdown cells compared to wildtype (Figure 5C). Secondly, knockdown of KDM5D resulted in a strong inhibition of osteogenic differentiation of male NCSCs, as shown by the reduced Alizarin Red S-stained calcium deposition after 30 days of culture on collagen type I fibers (Figure 5D). Quantification of the Alizarin Red Staining results confirmed the significant inhibition of male osteogenic differentiation upon siRNA-mediated knockdown of KDM5D (Figure 5E,F). In addition, we applied KDOAM-25 for the pharmacological inhibition of KDM5D in male NCSCs. After 30 days of culture on collagen type I fibers, we observed a significant reduction in calcium deposition in male NCSCs with increasing concentrations of KDOAM-25 (1 µM–100 µM) compared to untreated control (Figure 6A–C). These findings provide evidence of KDM5D being a novel crucial regulator of osteogenic differentiation in male craniofacial NCSCs.

## 4. Discussion

In this study, we demonstrated a decreased capacity of calvarial bone recovery in female rats as well as a cell autonomous delay in osteogenic differentiation of female human craniofacial stem cells compared to their male counterparts. Our findings revealed, in particular, a significantly decreased bone volume and bone mineral density in female rats compared to male animals upon transplantation of collagen fibers or Spongostan into critical-size calvarial defects. Accordingly, we previously reported the highly efficient closure of the calvarial lesions in male rats by collagen fibers or Spongostan [29] even in comparison to other clinically approved bone substitute materials such as NanoBone or Actifuse [30]. In line with our present observations, Meszaros and coworkers reported a significantly increased regeneration of cranial bone defects in male mice compared to their female counterparts after transplantation of gelatin sponges containing muscle-derived stem cells [14].

Transferring these promising findings to the human system, we further observed a strong delay in the osteogenic differentiation of adult human craniofacial neural crest-derived stem cells from female donors compared to those isolated from male ones. This observation is in accordance with the reduced capacity of female individuals for bone regeneration [3], reviewed in [4,5,6], and their increased risk of developing nonunion fractures [3,8] and elevated prevalence of osteoporosis [9,10,11]. Although certain osteogenic markers such as Runx2 or Osteonectin were not differentially expressed between male and female NCSCs, global transcriptional profiling revealed a broad range of differentially expressed genes between male and female NCSCs after 30 days of differentiation. Among others, these genes belonged to the KEGG pathways “ECM-receptor-interaction”, “cell adhesion molecules” or “focal adhesion”, the “TGF-beta signaling pathway”, the “MAPK signaling pathway” or the GO term “cell adhesion”. In line with our present findings, KEGG pathways such as “focal adhesion”, “ECM-receptor interaction” and the “MAPK signaling pathway” were previously reported to be enriched in human adult stem cells undergoing osteogenic differentiation compared to undifferentiated controls [39]. In particular, the “ECM-receptor interaction” pathway and expression of the gene ITGA1 can be directly linked to an interaction of NCSCs with the substrate consisting of collagen type I fibers harboring a nanoporous surface of 30 nm pores, in turn guiding their osteogenic differentiation [18].

Next to characteristic osteogenic markers such as BMP4 [40,41], MSX2 [42], COL4A4 and COL11A1 [43], we observed a range of osteogenic regulators strongly upregulated in male NCSCs after osteogenic differentiation compared to their female counterparts. Notably, we found autosomal genes such as C-FOS (chromosome 14), CYP7B1 (Cytochrome P450 Family 7 Subfamily B Member 1, chromosome 16) and FAM157C (family with sequence similarity 157, chromosome 8) among these regulators. Although C-FOS was described to determine the differentiation of murine hematopoietic cells into osteoclasts [44], it was likewise shown to precede osteogenic differentiation of cartilage cells [45]. Likewise, Ohta and colleagues reported fracture healing to be accompanied by an induction of C-FOS expression in osteoblasts within the ossifying callus [46]. Next to C-FOS, our present data revealed an elevated expression of CYP7B1 in male NCSCs after 30 days of osteogenic differentiation. CYP7B1 encodes a P450 enzyme, namely the oxysterol 7α-hydroxylase, which metabolizes the cholesterol metabolite 27-hydroxycholesterol (27HC), the first identified endogenous selective estrogen receptor modulator (reviewed in [47]). In patients, CYP7B1 mutations consequently led to an accumulation of circulating 27HC [48], which was in turn reported to inhibit the effects of estrogen [49]. Notably, CYP7B1 can be also considered as a major regulator of osteogenesis, as DuSell and coworkers demonstrated a KO of Cyp7b1 in female mice to result in decreased bone mineral density and bone formation rate compared to wildtype controls [50]. Likewise, we detected a male-specific elevated expression of the lncRNA FAM157C, which is commonly known to be expressed in the human bone marrow [51]. Li and coworkers showed deletion of FAM157C as one rare copy number variant being present in female patients suffering from turner syndrome [52]. Turner syndrome is a congenital disease caused by loss of one X-chromosome, which results, among other phenotypic characteristics, in a severe bone fragility including low bone mineral density and an elevated fracture risk (reviewed in [53]). In this line, the present observations reveal FAM157C as well as CYP7B1 and C-FOS as autosomal regulatory hallmarks of the osteogenic differentiation of male adult human stem cells compared to their female counterparts. Our findings may suggest the reduced expression of these regulators in differentiating female NCSCs as one underlying mechanism for the delay in osteogenic differentiation observed here.

Next to autosomal osteogenic regulators, the expression of genes such as USP9Y located on the Y-chromosome was significantly increased in male NCSCs compared to female NCSCs undergoing osteogenic differentiation. Interestingly, Shang and coworkers reported USP9Y to be alternatively spliced upon osteogenic differentiation [39], suggesting a regulatory role in osteogenic differentiation. We further particularly focused on one member of the Y-linked Lysin demethylase (KDM) family, namely KDM5D, which was highly upregulated in male NCSCs after 30 days of osteogenic differentiation compared to female NCSCs. Histone methylation is commonly known to be crucially involved in the osteogenic differentiation of stem cells, although the methylation status can either lead to the suppression or promotion of osteogenic differentiation (reviewed in [37]). For instance, the overexpression of KDM4A specific for H3K9me3 (trimethylated Lysin-9 at histone H3) enabled the osteoblast differentiation of mesenchymal stem cells [38]. Likewise, Yin and coworkers demonstrated the histone H3K4 trimethyltransferase Ash1l (absent, small, or homeotic-like) to promote osteogenic differentiation and osteogenesis [54]. On the contrary, KDM7A and KDM5A were reported to negatively regulate osteogenic differentiation of mesenchymal progenitor cells (KDM7A) or human adipose-derived stem cells (KDM5A) [55,56]. To the best of our knowledge, our present findings identify for the first time KDM5D, which demethylates H3K4me2/H3K4me1, as a critical driver of male osteogenic differentiation. Next to its elevated expression in male NCSCs after osteogenic differentiation, si-RNA-mediated knockdown or pharmacological inhibition of KDM5D significantly reduced the osteogenic differentiation capacity of male NCSCs. However, we cannot exclude the possibility that pharmacological inhibition of KDM5D at high dosage also acts on the viability of the cells. In view of our functional data, our findings demonstrate the expression of KDM5D as a master driver for the faster osteogenic differentiation of male NCSCs in comparison to NCSCs from female individuals. Future studies will investigate the changes driven by KDM5D in the epigenetic landscape of male NCSCs during osteogenic differentiation. In accordance with our findings, Shang and coworkers reported KDM5D to be alternatively spliced in human cartilage endplate-derived stem cells after osteogenic differentiation compared to controls [39].

While certain osteogenic regulators such as KDM5D, C-FOS, CYP7B1 and FAM157C were highly upregulated in male NCSCs after osteogenic differentiation, we observed a completely different expression pattern of osteogenic regulators in female NCSCs after the same exposure period to differentiation stimuli. Here, we found elevated transcript levels of X-linked pro-osteogenic genes such as KDM6A or XIST to be upregulated in female NCSCs compared to male ones. Hemming and coworkers reported the lysin demethylase KDM6A to promote osteogenic differentiation of MSCs in vitro and in vivo [57]. The long non-coding RNA (lnRNA) XIST was likewise shown to participate in the osteogenic differentiation of NC-derived periodontal ligament stem cells [58]. In line with the unchanged expression of osteogenic markers such as Osteonectin and Runx2 between male and female NCSCs, these findings indicate the at least partial osteogenic differentiation of female NCSCs after 30 days of cultivation on collagen type I fibers. However, we also observed negative regulators of osteogenic differentiation to be significantly more highly expressed in female NCSCs. These regulators particularly included the autosomally located gene MIR137HG (chromosome 1) as well as the X-linked gene TXLNG. Knockdown of MIR137HG was reported to promote the osteogenic differentiation of human adipose-derived stem cells [59]. The TXLNG gene encodes FIAT (factor-inhibiting activating transcription factor 4 (ATF4)-mediated transcription), the overexpression of which in mice was shown to lead to a reduced mineral deposition rate, in turn resulting in decreased bone mineral density and bone volume [60]. These observations are in line with our present observations linking an elevated expression of TXLNG to a delayed osteogenic differentiation and calcium deposition in female NCSCs. The expression of these negative osteogenic regulators in female NCSCs in combination with the decreased expression of major positive regulators such as KDM5D, C-FOS or CYP7B1 compared to their male counterparts may build the molecular basis of the observed delay in osteogenic differentiation in female craniofacial NCSCs.

In summary, our present findings show a decreased calvarial bone regeneration in female rats as well as a delayed osteogenic differentiation of female adult stem cells from the craniofacial region compared to their male counterparts. At the molecular level, this delay in osteogenic differentiation was accompanied by a sex-specific expression pattern of negative osteogenic regulators such as MIR137HG and TXLNG in female NCSCs, while male NCSCs revealed an elevated expression of pro-osteogenic regulators including C-FOS and CYP7B1. We further demonstrate the expression of Y-linked KDM5D as a prerequisite for the faster osteogenic differentiation of male NCSCs in comparison to NCSCs from female individuals. The present observation highlights the relevance of sexual dimorphisms in stem cell behavior for guiding bone recovery in clinical settings.

## Figures and Tables

**Figure 1 cells-11-00823-f001:**
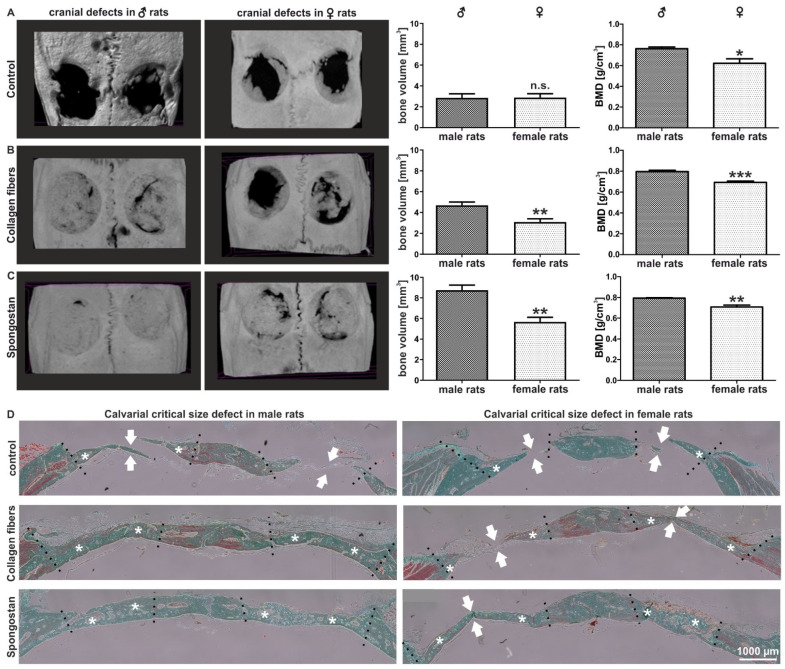
Female rats showed a reduced capacity for regenerating calvarial bone lesions after transplantation of Spongostan or collagen type I fibers into critical-size calvarial defects. (**A**) Micro-computed tomography (µCT) followed by quantification of bone volume and bone mineral density (BMD) revealed no closure of the critical-size defects in female or control male animals, but a significantly decreased BMD in female control rats. Mann–Whitney test, * *p* < 0.05, ** *p* < 0.01, *** *p* < 0.001 was considered significant. (**B,C**) µCT scans depicting a significantly decreased bone volume and BMD in female animals compared to their male counterparts after transplantation of collagen fibers or Spongostan. (**D**) Histological examination revealed elevated levels of newly formed bone in male animals upon transplantation of collagen fibers or Spongostan in male rats. Stars mark newly formed bone; arrows indicate unclosed lesions.

**Figure 2 cells-11-00823-f002:**
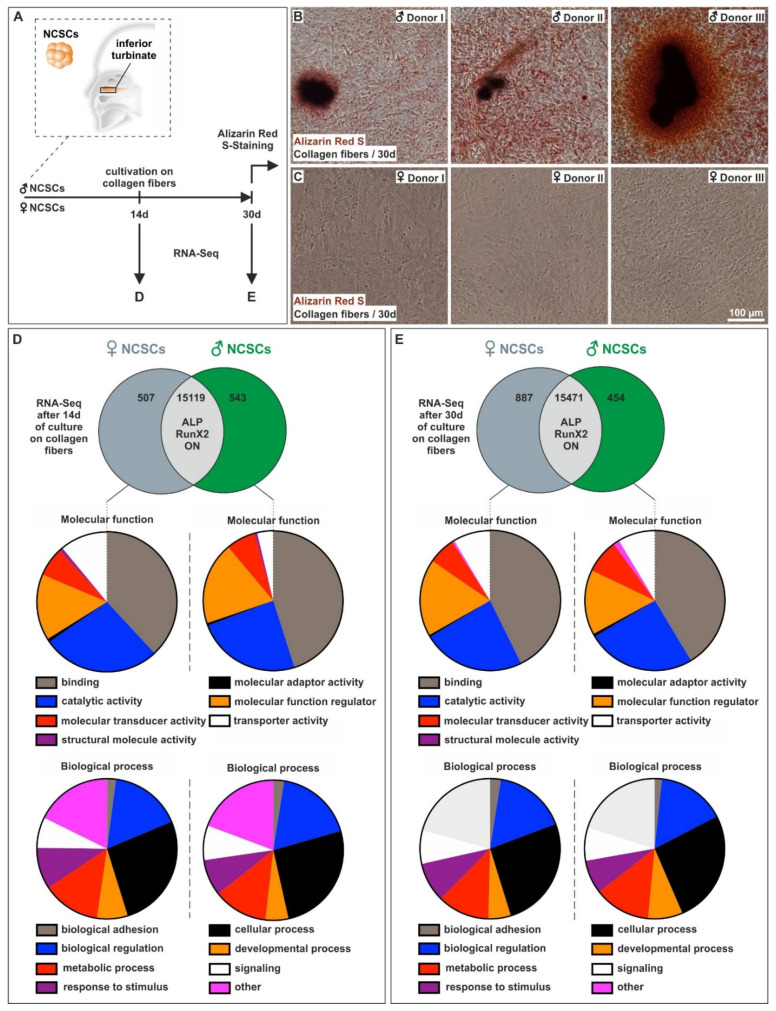
Adult human neural crest-derived stem cells from female donors showed a strong delay in osteogenic differentiation in comparison to male NCSCs. (**A**) Schematic depiction of the endogenous niche of craniofacial NCSCs as well as the experimental design. Scheme partly modified from [18]. (**B,C**) NCSCs from male individuals showed a strong Alizarin Red S-stained calcium deposition after culture on collagen fibers for 30 days, while no signs of Alizarin Red S-positive calcification were observable in female NCSCs. (**D,E**) RNA-sequencing revealed the expression of common osteogenic markers such as ALP, Runx2 and Osteonectin in NCSCs from three male and female donors after 14 and 30 days, while transcripts solely expressed in either male or female NCSCs belonged to similar GO terms.

**Figure 3 cells-11-00823-f003:**
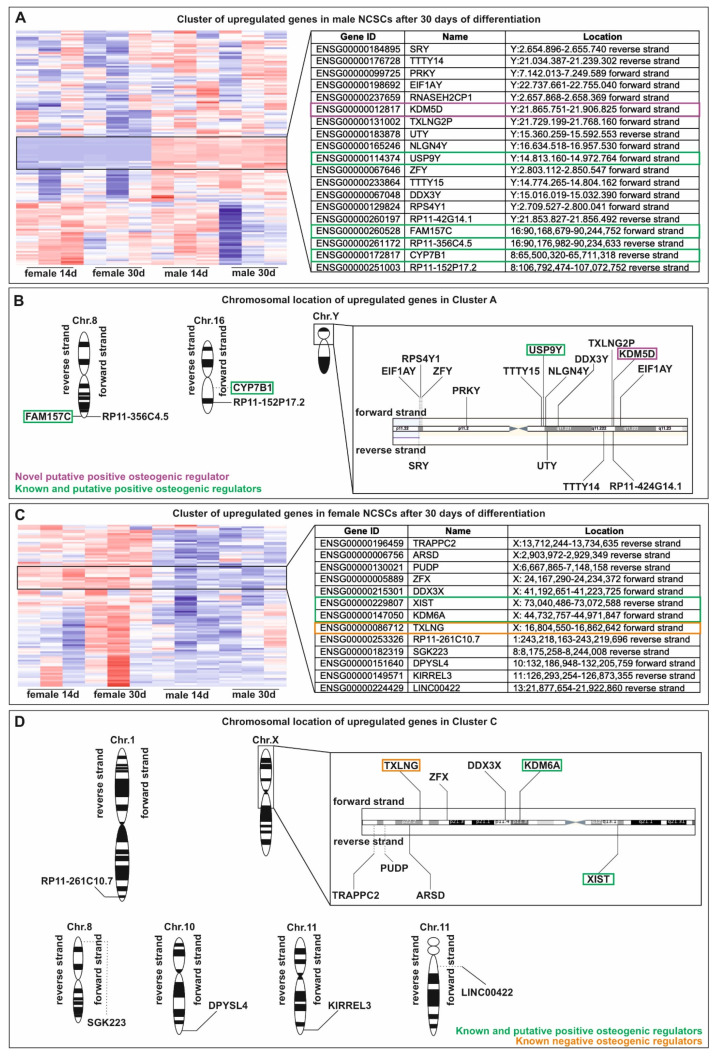
Hierarchical clustering of differentially expressed genes revealed a sexually dimorphic osteogenic regulator profile between male and female NCSCs undergoing differentiation. (**A,B**) Cluster of upregulated genes in male NCSCs comprising autosomal osteogenic regulator CYP7B1 as well as the Y-linked genes KDM5D and USP9Y. (**C,D**). Female NCSCs showed a cluster of upregulated genes including the pro-osteogenic regulators KDM6A and XIST as well as the negative regulator TXLNG.

**Figure 4 cells-11-00823-f004:**
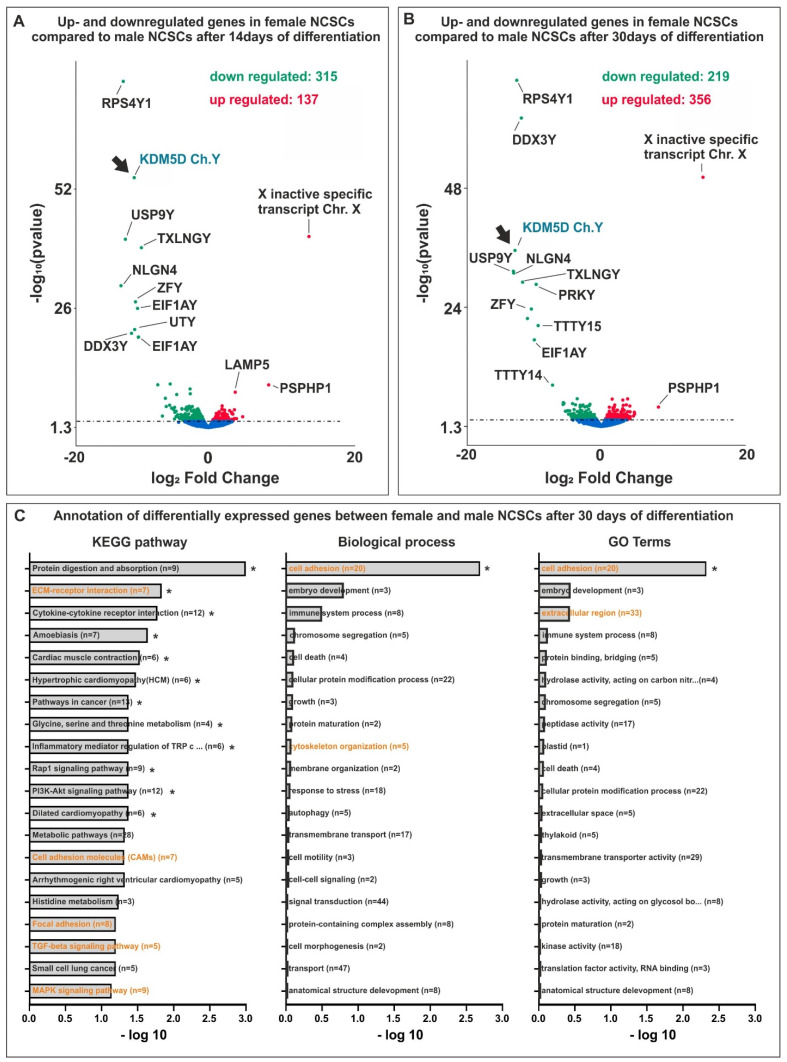
(**A,B**) Ribosomal Protein S4 Y-Linked 1 (RPS4Y1) and KDM5D (depicted in cyan, arrows) were most significantly downregulated in female NCSCs after 14 days and 30 of differentiation compared their male counterparts. (**C**) Differentially expressed genes between male and female NCSCs belonged, among others, to the osteogenic-related KEGG pathways and GO terms (depicted in orange). * *padj* < 0.05 was considered significant.

**Figure 5 cells-11-00823-f005:**
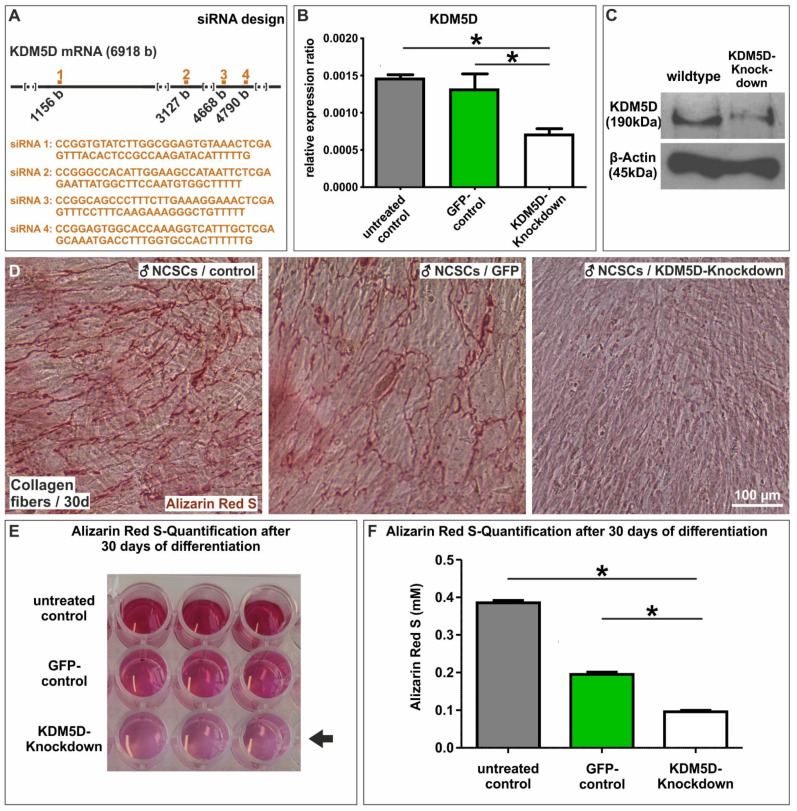
Knockdown of KDM5 significantly impaired the osteogenic differentiation capacity of male NCSCs. (**A**) Schematic view of the siRNA design. (**B**) qPCR analysis validating the knockdown of KDM5D. (**C**) Validation of the knockdown of KDM5D at the protein level via Western Blot. (**C**–**E**) Knockdown of KDM5D strongly inhibited the capability of male NCSCs to deposit calcium after 30 days of culture on collagen type I fibers compared to untreated and GFP-transduced cells. (**D–F**) Quantification of Alizarin Red S confirmed the significant inhibition of osteogenic differentiation in male NCSCs after knockdown of KDM5D. Mann–Whitney test, * *p* < 0.05 was considered significant.

**Figure 6 cells-11-00823-f006:**
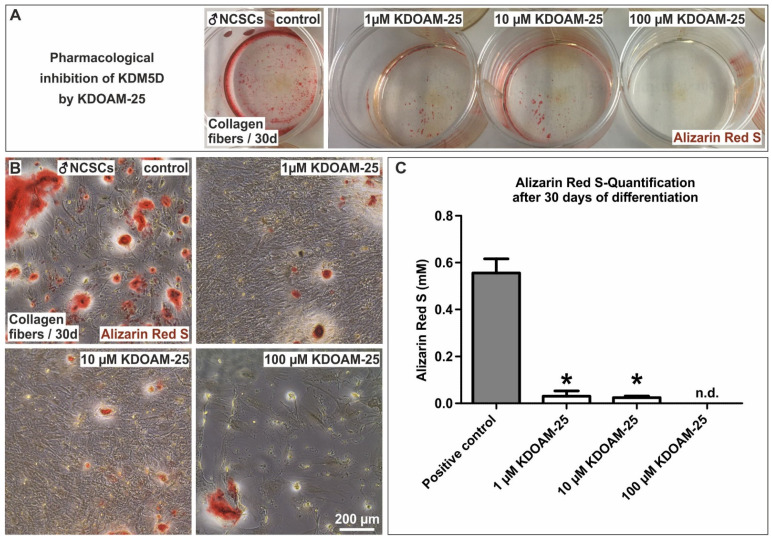
Pharmacological inhibition of KDM5D significantly reduced the capability of male NCSCs to undergo osteogenic differentiation. (**A,B**) Exposure of male NCSCs to increasing concentrations (1 µM–100 µM) of KDOAM-25 resulted in a significant reduction in calcium deposition after 30 days of differentiation compared to untreated control. (**C**) Quantification of Alizarin Red S confirmed the significant inhibition of male osteogenic differentiation upon inhibition of KDM5D by KDOAM-25-treatment. Mann–Whitney test, * *p* < 0.05 was considered significant. n.d.: not detectable.

**Table 1 cells-11-00823-t001:** Ages and passages of human NCSC donors.

Female Donors	Donor Age (y)	Passage of NCSCs
I	52	4
II	26	4
III	20	4
Male Donors		
I	26	3
II	44	4
III	22	4

**Table 2 cells-11-00823-t002:** Differentially expressed pro- and anti-osteogenic genes not included in clusters (Figure 3) between female and male NCSCs after 30 days of differentiation.

Gene Name	ENS ID	Down-/Upregulation in Female NCSCs Compared to Male NCSCs (log2 Fold Change)	*p*-Value
BMP4	ENSG00000125378	−1.2	0.014
MSX2	ENSG00000120149	−0.8	0.013
COL11A1	ENSG00000060718	−2.8	0.015
COL4A4	ENSG00000081052	−1.2	0.043
ITGA1	ENSG00000213949	−0.8	0.021
FOS	ENSG00000170345	−1.5	0.000
BMP7	ENSG00000101144	2.6	0.008
MIR137HG	ENSG00000225206	1.0	0.037

## Data Availability

The RNA-Seq datasets generated and analyzed during the current study will be available in the NCBI’s Gene Expression Omnibus repository [61] through the GEO Series accession number GSE185394.

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
