# Peer review of "Human Sex Matters: Y-Linked Lysine Demethylase 5D Drives Accelerated Male Craniofacial Osteogenic Differentiation"

_cells, 2022, doi:10.3390/cells11050823_

Round 1

Reviewer 1 Report

This article discusses the differences in osteogenesis ability between male and female and reveals the role of KDM5D in this process. But animal experiments seem to show that female rats had better bone regeneration in control group(Figure 1A). In addition, the relationship between animal experiments and cell experiments is not strong enough.

Reviewer 2 Report

This work by Merten et al addresses a very interesting issue that has been long overlooked. The epigenetic factors related to the different incidence of Osteoporososis in both sexes. Although this research would be, in principle, attractive to other scientists in the field, there are major problems that preclude its publication in its current form.

NCSCs seem like an odd choice to perform this study, since those cells need to be “encouraged” to osteogenically differentiate. Why did the authors choose this type of cells?. Using a cell type with intrinsic osteogenic capacity such as ASCs or BM-MSCs would have been much more efficient(?). Besides, if similar results were obtained using a model based on BM-MSCs would underscore the importance of their finding for the understanding of systemic bone loss associated pathologies such as osteoporosis. The authors need to make a a case for their choice and clearly explain their reasons.

Some technical details of the experimental groups are poorly defined. The authors need to clearly state what was introduced in the calvarial defect in each of the groups. Did the control group have no filling at all? It is slightly surprising that some closure of the defect can be clearly seen in the female control group. How can the authors explain this?

Since age of the donor has been repeatedly shown to greatly influence osteogenic differentiation it is key to give details about the age of the three donors used in their  in vitro osteogenic differentiation experiments. Also, since osteogenic potential seems to clearly decline after estrogen production decreases, these data is essential to know if the effect seen by the authors is purely linked to sex or the cells used have already suffered some changes in response to changes in the estrogen levels. Also, three samples seem a very little number of samples to work with and this number should be at least doubled.

How many samples were used for the transcriptome analysis? This is key information that has not been stated in the manuscript. It is also quite puzzling that the transcriptome analysis was done in cells already undergoing osteogenic differentiation.  How is this informative, since the authors have already seen that female cells have difficulties for undergoing differentiation? Shouldn’t the authors look for the root of the problem at early days when cell fate is determined? More importantly, none of the genes pinpointed as differentially expressed were actually measured in the cells. That is, no validation of the expression results is shown by the authors.

Also, more than 30 days seems like a very long time to achieve osteogenic differentiation in vitro since naturally bone-forming stem cells need lees than 30 day to produce a mineralized matrix, even when those cells have been sourced from an osteoporotic patient and have a clearly diminished osteogenic capacity. This brings to the table again the suitability of NCSCs for this kind of assays. In this regard, the title of the manuscript would be more accurate if it´d reflect this point.

Finally, KDM5D, as an epigenetic modulator, would clearly have a pleiotropic effect, changing the methylation state at thousands of different sites in the genome and thus, affecting the expression of multiple genes. Thus it is very unlikely to not have an effect on ostegenic differentiation when the expression of this protein is obliterated.  If KDM5D is in fact the motor of this change, the authors need to probe that the genes that they claim are linked to the different osteogenic capacity of male and female show an alteration of their key epigenetic marks at those osteogenic genes that would result in the reported differences in gene expression levels.

Also, although an effect of the pharmacological inhibition of KDM5D is clearly seen in figure 6, there is not proof that the inhibition is acting on osteogenic differentiation and not in the viability of the cells.

Reviewer 3 Report

The manuscript of Merten et.al is focused on understanding of possible cell-intrinsic mechanisms, which are responsible for sexual dimorphism in bone fracture healing. The authors employ in vivo rat model to proof this dimorphism and human in vitro model to investigate the mechanism. The topic of the study is interesting and important, the study design is relevant and findings look promising. Unfortunately, the description of material and methods part make it impossible to review the validity of the findings, as the statistical analysis part is fully missing. Besides this, many details and information are missing in the description of material and methods. Altogether, it makes this manuscript inappropriate for the publication and need the major review.

It is not clear, the RNAs from how many donors was used for RNA sequencing analysis (was one female vs one male)? If only one donor was used, how authors excluded intra-individual difference? How many reads was performed and which statistical analysis was applied? How the results of bioinformatic analysis were validated, with material of how many donors? Was the sequencing data also deposited in any data repositories in accordance with good scientific practice?

There is no information whether human donors (male and female) were matched by age?

Which primers (sequence) were used in qPCR? How many times the experiments were repeated and with how many replicates? How gene expression was normalized and calculated?

In addition, in description of methods the authors often refer to previous study, that in some cases make it difficult to find the real protocol used (due to another referencing to another reference).

In description of NCSC cell culture authors refer to the study (23) of Greiner, J.F et al, which compare two methods of NCSC culture. Which of them was used? Even common well known methods do not need detailed description, the project-specific details, such as cell number/passage, number of biological and technical replicates etc. must be provided.

Figures 1D and 2B are missing the scale bar.

Figure 1 A and 1 D: it looks like female animal has more bone tissue inside defect as male one (on both, µCT and histological images) in control group. How it could be explained? Did authors quantified a newly formed bone on histological samples, as this method is usually more sensitive as µCT?

Round 2

Reviewer 3 Report

The authors have significantly improved the quality of the manuscript and answered my comments and critic points. I do not have any farther concerns and find the manuscript suitable for publication.

Author Response

We gratefully thank the reviewer for finding our revised manuscript version suitable for publication.